# A Potential Alternative Orodispersible Formulation to Prednisolone Sodium Phosphate Orally Disintegrating Tablets

**DOI:** 10.3390/pharmaceutics13010120

**Published:** 2021-01-19

**Authors:** Essam A. Tawfik, Mariagiovanna Scarpa, Hend E. Abdelhakim, Haitham A. Bukhary, Duncan Q. M. Craig, Susan A. Barker, Mine Orlu

**Affiliations:** 1National Center for Pharmaceutical Technology, Life Science and Environment Research Institute, King Abdulaziz City for Science and Technology, P.O. Box 6086, Riyadh 11442, Saudi Arabia; 2Department of Pharmaceutics, UCL School of Pharmacy, University College London, 29-39 Brunswick Square, London WC1N 1AX, UK; scarpa.mariagiovanna@gmail.com (M.S.); hend.abdelhakim.16@ucl.ac.uk (H.E.A.); habukhary@uqu.edu.sa (H.A.B.); duncan.craig@ucl.ac.uk (D.Q.M.C.); m.orlu@ucl.ac.uk (M.O.); 3Department of Pharmaceutics, College of Pharmacy, Umm Al-Qura University, Makkah 24381, Saudi Arabia; 4Medway School of Pharmacy, The Universities of Greenwich and Kent at Medway, Anson Building Central Avenue, Chatham, Kent ME4 4TB, UK; s.barker@greenwich.ac.uk

**Keywords:** electrospinning, electrospun nanofibers, solvent-casting, orodispersible films (ODFs), prednisolone, disintegration, dissolution, e-tongue

## Abstract

The orally disintegrating tablet (ODT) has shown vast potential as an alternative oral dosage form to conventional tablets wherein they can disintegrate rapidly (≤30 s) upon contact with saliva fluid and should have an acceptable mouthfeel as long as their weight doesn’t exceed 500 mg. However, owing to the bitterness of several active ingredients, there is a need to find a suitable alternative to ODTs that maintains their features and can be taste-masked more simply and inexpensively. Therefore, electrospun nanofibers and solvent-cast oral dispersible films (ODFs) are used in this study as potential OD formulations for prednisolone sodium phosphate (PSP) that is commercially available as ODTs. The encapsulation efficiency (EE%) of the ODFs was higher (≈100%) compared to the nanofibers (≈87%), while the disintegration time was considerably faster for the electrospun nanofibers (≈30 s) than the solvent-cast ODFs (≈700 s). Hence, accelerated release rate of PSP from the nanofibers was obtained, due to their higher surface area and characteristic surface morphology that permitted higher wettability and thus, faster erosion. Taste-assessment study using the electronic-tongue quantified the bitterness threshold of the drug and its aversiveness concentration (2.79 mM). Therefore, a taste-masking strategy would be useful when further formulating PSP as an OD formulation.

## 1. Introduction

Oral drug delivery is a preferable route of drug administration due to high patient compliance and ease of administration [1]. One particular oral dosage form, which showed a considerable potential in the last two decades is an orally disintegrating tablet (ODT), owing to its acceptability and safety profiles compared to the conventional tablets [2]. Once it is in contact with saliva, it will rapidly disintegrate and release its active drug allowing for its swallowing without the need for additional liquid [3,4]. This dosage form is beneficial with specific populations who have difficulties in swallowing, such as pediatrics, geriatrics, and bedridden patients [5,6,7]. According to the United States Food and Drug Administration (US FDA), ODTs should offer a fast disintegration time (≤30 s) and a tablet weight of ≤500 mg [8]. These are considered to be the two most vital features of this oral dosage form and have created some challenges for the formulator [2]. Along with their ease of swallowing, high dosing accuracy and cost-effectiveness, OD formulations can also have additional benefits compared to conventional formulations. Rapid onset of action and avoiding first-pass metabolism can be advantageous, in which the overall dose of the active ingredient can be reduced, resulting in a formulation with potentially fewer side effects and ultimately better patient compliance and outcomes [9,10].

However, because of the bitterness of many active ingredients, patient compliance may be affected, leading the pharmaceutical industry to research new taste-masking technologies that would enhance the palatability of ODTs. Among these techniques, the most straightforward popular approach is the use of flavors or sweetening agents, yet, this approach may not be effective in the presence of a very bitter drug or a drug that is administrated at high doses [9]. Other taste-masking techniques include complexation or coating with taste barrier polymers, such as Eudragit^®^ E PO. The addition of these polymers can increase the weight of the ODT, which may lead to an unacceptable mouthfeel such as grittiness, particularly in high-dose drugs [11]. ODTs are sensitive to humidity due to the use of excipients that are moisture-sensitive, hence, superior protection packs are required. Depending on the process of ODT preparation, some of these techniques could produce fragile tables, such as those prepared by lyophilization [11].

A study by Bright (2014) explored the taste-masking of the extremely bitter drug dextromethorphan hydrobromide using complexation methods, such as ion-exchange resin complexation and inclusion complexation using cyclodextrin. The results indicated that the former process could mask the bitter taste of the drug more efficiently [12]. Another study, by Hesari et al. (2016), explored the taste-masking of ranitidine by using a sweetening agent, granulation, solid dispersion with soluble and insoluble agents, and complexation with cellulose derivatives. The latter method was reported to be the most successful in terms of achieving a very rapid disintegration time (around 5 s) and enhanced taste-masking ability [13]. Therefore, depending on the drug and formulation in question, the use of taste-masking techniques such as the complexation method has proved successful in a number of highly bitter drugs. For orodispersible formulations, finding an alternative orodispersible formulation that is taste-masked and capable of maintaining the rapid disintegration time (≤30 s) of ODTs, in addition to, not exceeding the recommended tablet size (i.e., 500 mg) is required.

In this study, electrospun fibers and orally disintegrating films (ODFs) were chosen as potential alternatives to ODTs, in which prednisolone sodium phosphate (PSP) was the model drug. This is due to its availability as ODT (Orapred ODT^®^) and to its various therapeutic indications, in both geriatric and pediatric population, to treat several health conditions such as severe allergies, breathing difficulties (e.g., asthma and croup), immune system disorders (e.g., systemic lupus erythematosus), hematological disorders (e.g., hemolytic anemia and chronic lymphocytic leukemia), and rheumatoid arthritis [14].Furthermore, PSP was also suitable to be loaded onto multiple dosage forms. It was used in the salt form due to its improved solubility and therefore, oral bioavailability [15]. PSP requires dosages starting from 1 mg/day up to a maximum dose of 60 mg/day in children [14], and fast disintegrating oral drug delivery systems show promise as age-appropriate formulation platforms.

ODFs are composed mainly of film-forming polymers, which can be either natural or synthetic. Depending on the properties of these polymers, such as solubility and molecule weight, different drug loading, release rate, disintegration profile, and mechanical properties can be achieved [16]. Despite the simplicity of preparing ODFs, there are many reported limitations accompanied with this dosage form, such as the difficulty of attaining dose uniformity and even thickness of the formed films, which might affect the mechanical properties and release behavior of the films within the same batch [17]. Owing to this, an alternative system which showed huge potential in controlling the release rate and having unique mechanical properties, are electrospun nanofibers [18]. Electrospinning mainly uses electrical forces to overcome the surface tension of a viscous polymer solution continuously forming a jet, and thus, producing a fibrous mat upon solvent evaporation. Electrospun fiber mats have gained widespread interest in the field of drug delivery, particularly wound healing and tissue regeneration, due to their structural flexibility, tensile strength, high surface area to volume ratio, resemblance to the extracellular matrix proteins, and ability to entrap water-soluble and -insoluble drugs by using a wide range of natural- or synthetic-occurring polymers [19,20]. Besides, it has also been used to fabricate fiber-mats intended for use in children, via the use of taste-masking polymers [20]. Coaxial electrospinning offers the ability to prepare a core and shell nanofibers to either loaded multiple drugs or to coat the fibers with a different polymer to obtain different behavior. For instance, to coat a hydrophilic polymeric fiber with a layer of hydrophobic polymer to delay the release of the drug located in the core layer [19]. In order to achieve fibers with desired properties and functions, solution and process parameters need to be optimized for the purpose. The polymer solution viscosity and conductivity play essential roles in stabilizing the jet along with the process’s flow rate, applied voltage, and the gap distance between the nozzle and the collector plate [21]. When optimized accordingly, the fiber can be fabricated to into highly porous structures exhibiting a higher surface area and therefore, permitting quicker diffusion of the drug from the polymer matrix [22].

Polyvinyl alcohol (PVA), a film-forming polymer, has been widely used in both medical and food industries since 1930 [23]. It is a semi-crystalline, water-soluble, synthetic polymer that can undergo degradation by a combination of oxidase and hydrolase enzymes into acetic acid and has been approved by the US FDA for human consumption, and is considered generally regarded as safe (GRAS) [23]. PVA has been incorporated into the manufacture of many medical products such as sutures, transdermal patches, tablet formulations, surgical devices, contact lenses, and artificial organs, due to its biodegradability, biocompatibility, non-toxicity, non-carcinogenicity, and film-forming ability [24,25,26]. Moreover, it has been used in artificial tears products, such as Hypotears^®^, Refresh^®^, and Liquifilm^®^, to treat dry eye symptoms due to its coating properties [27,28]. PVA is commonly used in the composition of both ODFs and electrospun fibers. Several reports have demonstrated the ability to solvent-cast and electrospin PVA to produce ODFs and orally administrated nanofibers, respectively [18,22,29,30,31].

In this study, PVA was used to prepare both the PSP-loaded electrospun fibers and ODF as potential alternatives to the marketed Orapred ODT^®^. The EE%, disintegration, and release behavior of both drug-loaded formulations will be compared along with the solid-state characterization. In addition, the taste of drugs needs to be considered for patient acceptability. PSP has been shown to have an unpleasant taste, an issue which may affect adherence [32,33]. Therefore, the bitterness threshold for this drug was quantified by an electronic tasting system, or e-tongue, to report the minimum concentration of PSP that is known to be aversive.

## 2. Materials and Methods

### 2.1. Materials

High molecular weight (MW) PVA 197,000 with a degree of hydrolysis of 85–89% and ethanol were obtained from Merck-Sigma Aldrich Company Ltd. (Darmstadt, Germany). PSP was purchased from Cambridge Bioscience Ltd., (Cambridge, UK). The chemical structure of PSP and PVA are presented in Figure 1. Quinine hydrochloride dihydrate, potassium chloride, tartaric acid, absolute ethanol, and hydrochloric acid (32%) were bought from Sigma-Aldrich (Dorset, UK), while distilled water was generated by an ELGA Option 4 Water Purifier (Veolia Water Technologies, High Wycombe, UK).

### 2.2. Methods

#### 2.2.1. Preparation of Electrospun Nanofibers

PVA nanofibers were prepared by modifying the electrospinning method of Supaphol and Chuangchote [34], using a Spraybase^®^ electrospinning instrument (Spraybase^®^, Dublin, Ireland). The PVA solution was prepared by dissolving PVA in 9 mL distilled water at 80 °C for one hour. After cooling, 1 mL of ethanol was added, and the mixture was stirred for 30 min to obtain a final volume of 10 mL and a final concentration of 6.5% *w*/*v*. Then, 1.3% *w/v* PSP, or a drug to polymer ratio of 1:5, was added and stirred for 30 min to obtain a homogenous solution. A stable jet was achieved after varying the flow rate and the applied voltage to be 0.8 mL/h and 18–20 kV, respectively. The needle diameter of 0.7 mm and a distance between the tip of the needle and the collector of 15 cm, were kept constant. The room temperature and relative humidity were 22–24 °C and 40–45%, respectively. The end product nanofibers were collected on aluminum foil and were then peeled off and stored in a sealed plastic container at ambient temperature for further testing.

#### 2.2.2. Preparation of Solvent-Cast ODFs

PVA films were prepared by modifying the solvent casting method of Birck et al. [35]. PVA solution was prepared similar to the nanofibers PVA solution for a final concentration of 6.5% *w*/*v*. Then 1.3% *w*/*v* PSP was added and stirred for 30 min. After reaching a homogenous solution, 7.5 mL was poured in an 8 cm diameter silicone mold (Shenzhen Yimeifen Technology, Guangdong, China) positioned on top of a polyvinyl chloride (PVC) sheet (Vesey Arts and Crafts, Sutton Coldfield, Birmingham, UK). The cast solution was then heated at 40 °C for one hour. The film was then peeled off, wrapped in aluminum foil and stored in a sealed plastic container at ambient temperature for further testing.

#### 2.2.3. Morphological Characteristics

The morphology of the prepared nanofibers and ODFs were examined using scanning electron microscopy (SEM). A 0.4 cm × 0.4 cm piece of foil on which the nanofibers were collected and an equivalent amount of ODF was adhered onto an SEM stub, using double-sided carbon tabs (Agar Scientific, Stansted, UK). The prepared stub was then given a thin coating of gold (10 nM) in a Quorum Q150T Sputter Coater (Quorum Technologies Ltd. East Sussex, UK) in an argon atmosphere. The coated stub was then transferred and imaged under FEI Quanta 200F (FEI company Ltd., Eindhoven, The Netherlands), at an acceleration voltage of 5 kV. Fiber size analysis was performed by measuring the diameter of at least 100 fibers using ImageJ software (National Institutes of Health, Bethesda, MD, USA).

#### 2.2.4. Thermal Analysis and Solid-State Characterization

Thermogravimetric analysis (TGA), X-ray diffraction (XRD), and Fourier Transform Infrared (FTIR) spectroscopy were performed on the raw materials, physical mixture (PM) of 6.5% PVA and 1.3% PSP, as well as, the blank and drug-loaded nanofibers and ODFs. The PM was prepared by gently mixing the drug and polymer powders using a mortar and pestle, then the mixed powder was transferred into an Eppendorf tube for a further mixing using vortex, in order to ensure a proper distribution of the drug within the polymer powder.

##### Thermogravimetric Analysis

TGA was performed using a TA Hi-Res TGA 2950 thermogravimetric analyzer (TA Instruments UK, Herts, UK). A purge nitrogen gas flow rate of 60 mL/min was used for the furnace and 40 mL/min for the TGA head. An aliquot sample (weight range from 5 to 10 mg) was placed into an open aluminum pan (PerkinElmer, Waltham, MA, USA). The samples were equilibrated at 30 °C and heated to 400 °C at a rate of 10 °C/min. Data were viewed on TA Universal Analysis software, version 4.5A and were plotted and analyzed using OriginPro 2016 (OriginLab Corporation, Northampton, MA, USA).

##### Fourier Transform Infrared

FTIR analyses were conducted using a Spectrum 100 FTIR spectrometer (PerkinElmer, Waltham, MA, USA). The samples were scanned over the range 4000–650 cm^−1^, with the spectral resolution set at 1 cm^−1^. An average of 4 scans of each sample were recorded. Background scans were performed in all experiments. Data were plotted and analyzed using OriginPro 2016 (OriginLab Corporation, Northampton, MA, USA).

##### X-ray Diffraction

X-ray diffraction patterns were obtained by a MiniFlex 600 diffractometer (RigaKu, Tokyo, Japan) supplied with Cu Kα radiation (λ = 1.5148 227 Å) at a voltage of 40 kV and a current of 15 mA. Samples were fixed on an aluminum plate, and analysis recorded over 2θ range between 3 and 45° (step size of 0.05° and time per step of 0.2 s). RAW files produced were converted to X-ray diffraction (XRD) data files using PowDLL version 2.51 file converter software. The data were then viewed on X’Pert Data Viewer version 1.2F and were plotted and analyzed using OriginPro 2016 (OriginLab Corporation, Northampton, MA, USA).

#### 2.2.5. Ultraviolet Assay for Drug Determination

In order to assess the drug loading and the release profile of the drug-loaded nanofibers and ODFs, PSP calibration curves were developed using absorbance data recorded using a Jenway 6305 UV–Vis spectrophotometer (Bibby Scientific, Staffordshire, UK). These were performed in distilled water and simulated saliva fluid (SSF), for drug loading and release profile determination, respectively. The SSF was prepared, according to Pinďáková et al. [36] at a pH of 6.8. A wavelength of 246 nm and a 1 mL cuvette were used for all calibration curves based on the Bhusnure et al. study [37]. A serial dilution was performed to obtain PSP solutions in the range of 75–0.39 µg/mL in distilled water and SSF.

#### 2.2.6. Drug Loading (DL) and Encapsulation Efficiency (EE%)

The DL and EE% assessments were measured by a modified version of Tawfik et al. [38]. Certain weights of the drug-loaded nanofibers and ODFs were dissolved in 40 mL of distilled water. After the complete dissolving, 1 mL of the formed solution was withdrawn, and the drug content was measured by using the calibration curve and calculated using the following equation:(1)$$\mathrm{DL}=\frac{\mathrm{Entrapped}\;\mathrm{Drug}\;\mathrm{Amount}}{\mathrm{Amount}\;\mathrm{of}\;\mathrm{yield}\;\mathrm{Formulations}}$$

The following equation calculated the EE%:(2)$$\mathrm{EE}\%=\;\frac{\mathrm{Entrapped}\;\mathrm{Drug}\;\mathrm{Amount}}{\mathrm{Total}\;\mathrm{Drug}\;\mathrm{Amount}}\times 100$$

The results represent the mean (±SD) of six replicates.

#### 2.2.7. Disintegration Test

The disintegration times of electrospun nanofibers and ODFs were measured by the modified petri dish method of Illangakoon et al. [39]. Square sections of the nanofibers and the ODFs that weighed 17 and 15 mg (equivalent to a therapeutic dose of 2.5 mg PSP), respectively, were placed into 5 mL of pre-warmed SSF (the disintegration medium), and the measurement was performed at 37 °C and under gentle stirring using shaking incubator. Time was measured until complete disintegration (i.e., full detaching of the sample) occurred. The results represent the mean (±SD) of three replicates.

#### 2.2.8. Drug Release Study

The release of the nanofibers and the ODFs were performed using SSF (pH 6.8) as the release medium. A certain number of nanofibers and ODFs that is equivalent to 2.5 mg of PSP were sunk using custom-made stainless-steel sinkers, as shown in Figure 2. The volume of the media used was 60 mL for each vial. Vials were kept in a shaking incubator at 37.2 °C and a 75 RPM shaking speed. The cumulative release percentages were measured as a function of time. This was calculated according to the following equation:(3)$$\mathrm{Cumulative}\;\mathrm{amount}\;\mathrm{of}\;\mathrm{release}\;\left(\%\right)=\frac{\mathrm{Ct}}{C\infty }\;\times 100$$
where (Ct) is the amount of PSP released at time (t) and (C∞) refers to the total amount of drug-loaded formulation. The results represent the mean (±SD) of three replicates.

#### 2.2.9. E-Tongue and Bitterness Threshold Determination

The TS-5000Z taste sensor (Intelligent Sensor Technology Inc., Atsugi, Japan) was used. The sensor AC0 (purchased from New Food Innovation Ltd., Nottingham, UK) that can detect basic bitter cationic substances was used to determine the bitterness intensity of raw PSP. Sensor checks were carried out before every measurement to ensure that the sensors were calibrated (i.e., ensure a correct mV range). Each sample was measured four times. The first run was discarded as recommended by the manufacturer to allow for sensor conditioning. Each measurement cycle consisted of measuring reference potential (V_r_) in a reference solution, followed by the measurement of electric potential (V_s_) of the sample solution; V_s_ − V_r_ represented the initial taste. The sensor was finally refreshed in alcohol solutions before measuring of the next sample.

The reference solution was prepared by dissolving 30 mM potassium chloride and 0.3 mM tartaric acid in distilled water. The negatively charged membrane washing alcohol solution was prepared by diluting absolute ethanol to 30% *v*/*v* with distilled water, followed by the addition of 100 mM hydrochloric acid. This method is adapted from Abdelhakim et al. [20].

The bitterness threshold of PSP was calculated as a comparison to quinine HCl dihydrate, a commonly used bitter standard drug with known bitterness and aversiveness levels in humans. In human taste panels, bitterness thresholds are determined by selecting the concentration of the drug that produces half of the maximal rating (100), known as the EC50. For the e-tongue, bitterness thresholds are deduced by using the human EC50 for quinine HCl dihydrate and finding the corresponding mean sensor response at that value. For quinine HCl dihydrate, this value is known to be 0.26 mM [40].

#### 2.2.10. Statistical Analysis

The regression equation, correlation coefficient, mean and standard deviations were calculated using OriginPro 2016 software (OriginLab Corporation, Northampton, MA, USA). For the release study, the mean comparison was performed by parametric T-test using GraphPad Prism^®^ statistical software (GraphPad Software, San Diego, CA, USA). The *p* < 0.0001 was taken as a criterion for a statistically significant difference.

## 3. Results and Discussion

### 3.1. Morphological Characteristics

The morphology of the drug-loaded electrospun nanofibers and ODFs were examined under SEM, as shown in Figure 3. The drug-loaded solvent-cast ODF showed no drug crystals on its surface, indicating that the drug is incorporated within the film (Figure 3A). On the other hand, owing to the high MW of PVA (197,000 Dalton) in the formulations, flattened fibers were obtained, as illustrated in Figure 3B. It was reported in Koski et al. study [41] that by increasing the MW of PVA, more flattened fibers were obtained due to the reduction in the rate of solvent evaporation as the MW is enhanced, hence, wet fibers were produced which flattened upon their impact with the collector. Therefore, 10% *v*/*v* ethanol was added to the polymer solution, in order to overcome these ribbon-like shape fibers, as can be seen in Figure 3C. The surface of these drug-loaded nanofibers was smooth and un-beaded with a mean diameter of 257 ± 52 nm. Ethanol has reduced the voltage that was used, increased the evaporation rate of the PVA solution, and decreased the conductivity and surface tension of the water that allowed it to overcome the production of wet fibers (i.e., entrapped solvent), as reported in the Sukyte et al. [42] and Asawahame et al. [43] studies. The nanofibers’ size distribution is exhibited in Figure 3D. Moreover, there was no evidence of drug crystals on the surface of these fibers when a drug to polymer ratio of 1:5 was used.

Furthermore, the choice of a proper PVA degree of hydrolysis (DH) was crucial in this study. It was demonstrated by Zhang et al. [44] that nanofibers prepared from three different DH (80%, 88%, and 99%) influenced their morphology. They reported that PVA with 80% DH resulted in a ribbon-like shape fibers due to lack of dryness of these fibers at the time of collection and the PVA solution was the highest in viscosity. Nanofibers made from 88% DH showed the most uniform morphology. Besides, smaller diameter fibers were produced due to the relative lower viscosity of this PVA solution compared to the other 80% and 99% DH PVA. The electrospun fibers made from 99% DH were beaded. Furthermore, it has been shown that by using a higher MW of PVA and 99% DH was challenging to spin. Therefore, the 88% DH was used in this study.

### 3.2. Thermogravimetric Analysis

TGA analysis was performed to detect any residual solvent, i.e., entrapped water in the case of raw materials and ethanol in the case of the nanofibers’ formulations. Ethanol was added to the electrospinning solution to stabilize the jet, as mentioned above.

As shown in Figure 4, PSP exhibited an initial weight loss culminating at 100 °C that could be attributed to the presence of moisture in the powder product of this drug. The degradation of PSP started around 250 °C, and it occurred in two phases as it was also reported in a previous study by ElShaer et al. [45]. PVA showed a slight weight loss (7%) after 100 °C which accounts for the evaporation of the moisture, owing to the hygroscopic property of this polymer. PVA degradation onset started around 275 °C that is consistent with Yan et al. [46]. In comparison with pure PVA, the degradation onset of the PM was slightly shifted towards lower temperatures due to the presence of small amounts of PSP.

Drug-loaded and blank ODFs showed a 7% weight loss after 100 °C, due to the evaporation of the remaining water. A weight loss of around 5% was noticeable in both drug-loaded and blank nanofibers at 70 °C, whereas no further weight loss was observed until degradation. This weight loss might indicate the remaining portion of ethanol, which has a boiling point of about 78 °C. Another explanation for this lower solvent evaporation temperature could be related to the wettability of the fiber-mat. Wettability determines the degree at which a liquid spreads on a solid material [47]. The combination of high wettability of low DH PVA [48] and the increased surface area of electrospun nanofibers [49,50] could potentially justify the fast uptake and release of moisture. Therefore, in comparison to films, where the polymeric chains are tightly packed, and the total surface area exposed to air is smaller, electrospun nanofibers might exhibit faster solvent evaporation rates [51]. The same degradation points were observed for blank and drug-loaded films initiating at 275 °C, in accordance with the degradation onset observed for PVA. A weight loss of 50% was reached at 341.6 °C, and 346.4 °C in blank and drug-loaded nanofibers, respectively, and at 336.5 °C and 347.9 °C in blank and drug-loaded ODFs, respectively.

This result suggested that the nanofibers might contain a slight residual amount of ethanol.

### 3.3. Fourier Transform Infrared

The compatibility between drug and polymer in the formed formulations is an important factor to avoid solid phase separation over time. FTIR analysis was conducted to assess the drug–polymer molecular interactions. The chemical structure of PSP and PVA are presented in Figure 1. The FTIR spectrum of the pure PSP, PVA, and their PM exhibited the characteristic peaks of each pure raw material, as shown in Figure 5. The broad peak between 3000 and 3700 cm^−1^ that appeared in all spectra, is due to H-bond stretching vibrations of O–H group, whereas, the peak that presented between 2950 and 2800 cm^−1^ represent the C–H and C=H stretching. In the fingerprint region, PSP distinctive peaks were shown at 1710 and 1655 cm^−1^ (C=O stretching), 1596 cm^−1^ (aromatic C–C bending) and 1014 cm^−1^, while PVA peaks appeared at 1150 to 1090 cm^−1^ (C–O stretching). The spectrums of both drug-loaded nanofiber and ODF formulations demonstrated an overall decrease in their intensities with the distinctive drug peak at 1596 cm^−1^ (aromatic C–C bending) was observed, while it was absent in the blank formulations, as shown in Figure 5.

This finding indicated the presence of the drug (PSP) within the nanofibers and ODFs and the lack of chemical interactions between the drug and the polymer that might occur due to the manufacturing processes. Therefore, this suggests the absence of any changes in the drug and polymer structural integrity, as well as, any loss of the efficiency of the drug, which is consistent with the Palanisamy and Khanam study [52] on prednisolone in various water-soluble carriers.

### 3.4. X-ray Diffraction

XRD analysis was conducted for the solid dispersion detection of the PSP in each formulation in comparison to the raw materials. Diffractogram results are shown in Figure 6. The characteristic peaks of PSP appeared between 3° and 5° indicated the presence of the drug in the crystalline form. For the raw PVA, the distinctive peaks between 18° and 21° showed its crystallinity [53]. Reflections of both characteristics’ peaks appeared at the PM. In the blank formulations, the characteristics reflections of crystalline PVA seemed more intense in the solvent-cast ODF compared to nanofibers that show more conversion of the crystalline structure into semi crystal to amorphous form (i.e., broad halo). The drug-loaded formulations appeared as broad halos and lacked the distinctive peaks of the drug (i.e., 3–5°) owing to the molecular dispersion transformation of both formulations. This molecular dispersion was also demonstrated in Illangakoon et al. paracetamol/caffeine-loaded nanofibers and ODFs [39], and Asawahame et al. propolis-loaded nanofibers [43].

### 3.5. Ultraviolet Assay for Drug Determination

The UV assays were simple and rapid, in which PSP was determined in both distilled water and SSF to measure the drug loading and the release profile of the drug-loaded formulations, respectively. PSP calibration curves showed significant intra- and inter-day precisions. The drug calibration curves (presented in the Appendix A) in distilled water and SSF demonstrated excellent linearity at a concentration range of 75 to 0.39 µg/mL with the regression equation and the correlation coefficient (r^2^) of y = 0.02653x + 0.00287 (r^2^ = 0.9999) and y = 0.02704x − 0.00632 (r^2^ = 0.9991), respectively.

### 3.6. Drug Loading (DL) and Encapsulation Efficiency (EE%)

The EE% of solvent-cast ODFs was higher (~100%) compared to the nanofibers (~87%), as shown in Table 1. Previous studies were able to cast films with almost 100% EE [54,55,56]. However, Taepaiboon et al. [22] and Thitiwongsawet and Supaphol [30] studies showed an incomplete encapsulation of their PVA fibers which was reported to be above 80%. This difference in the EE% might be attributed to the mechanical processing faults, which would require further investigation. The DL was calculated as approximately 146 and 167 µg/mg for the drug-loaded nanofibers and ODFs, respectively. This result was considered later in the release study, to test the release profile of both formulations in equivalent DL. It is worth mentioning that in order to reach a therapeutic dose of PSP ODT, i.e., 13.4 mg, approximate weights of 92 and 80 mg of the drug-loaded nanofibers and ODFs, respectively, will be sufficient to deliver this required dose compared to the 500 mg tablet weight of a single ODT.

### 3.7. Disintegration Test

The disintegration time of drug-loaded nanofibers and ODFs, with an average weight of 17 ± 1 and 16.5 ± 1 mg, respectively, was measured. The results showed that the average disintegration time of the drug-loaded nanofibers was substantially faster (30 s) than the ODFs (≈700 s). As observed by SEM image (Figure 3B), the nanofibers exhibited a less compact matrix than the films, allowing air pockets to form within the structure and favoring a faster adsorption of liquid by capillarity [57]. The characteristic surface morphology confers the nanofibers high wettability [18]. Moreover, the nanofibers showed a large surface-area-to-volume ratio, thus, exposing an overall larger surface area than the ODFs to hydrate and therefore, to erode [58]. This finding can be an advantage for the nanofibers, in which a faster drug release could be achieved due to the fast disintegration behavior of this formulation.

### 3.8. Drug Release Study

Owing to lack of official pharmacopeia drug release guidelines for the nanofibers and ODFs, the release experiment was designed based on the nature of both formulations and the route of their administration. Increasing the shaking speed to more than 100 RPM will allow a quicker drug release profile, while using a lower rate, less than 50 RPM, will slow the release. Therefore, a medium-range speed of 75 RPM was used. This observation was consistent with Scheubel et al. study, who experimented the effect of changing the paddle speed on prednisone tablets using small vessel USP apparatus [59]. 60 mL of release medium (i.e., SSF pH 6.8) was used, to fit the release study to the sink condition (i.e., 3 to 10 times more than the saturated drug solubility). Furthermore, due to the floating characteristic of the nanofibers and ODFs, custom-made sinkers were used to ensure that the formulations were kept at the bottom of the vessels during the release study, as it was suggested by Tawfik et al. [38]. This release study was performed up to 4 h until the full release of the loaded-drug was obtained.

The drug release rate of the nanofibers was found to be significantly faster (*p* < 0.0001) than the ODFs, as shown in Figure 7. The release from the nanofibers showed that 24% of the drug was released in the first 5 min compared to 13% liberated from the films. After an hour, the cumulative release was 91% and 77% for the nanofibers and films, respectively. Owing to the fast disintegration of the nanofibers, the release of PSP was found to be quicker. A similar observation was shown in the release studies of Li et al. [18] on PVA nanofibers loaded with caffeine or riboflavin compared to cast film formulations; Nagy et al. [29] on Donepezil salt loaded in PVA fibers compared to the cast film, and Thitiwongsawet and Supaphol [30] on ultrafine PVA fibers loaded with carbendazim compared to the film. However, in solvent-cast films, the salt form of a drug is usually released in a diffusion-controlled profile, as explained in Li et al. study [18]. Also, due to the large water retention, the high degree of swelling and the larger surface area of the nanofibers compared to the films, a faster drug release profile was achieved. This finding was also concluded in Thitiwongsawet and Supaphol study [30]. All the previous release studies have demonstrated the enhanced dissolution rate of electrospun fibers compared to solvent-cast films due to the slower film disintegration and slower diffusion of the medium. This result was also observed in Krstić et al. [60] on the hydrophobic drug ‘carvedilol’ that was loaded into PEO electrospun nanofibers and solvent-cast films. On the other hand, the dissolution of PSP ODT (Orapred ODT^®^) was previously tested by Adinarayana et al. [61] in three different release media, i.e., water, 0.1N HCl and acetate buffer (pH 4.5). The results showed that 81% of the drug was released after 45 min in water, while full PSP release was obtained at 30 min in the acetate buffer and after 60 min in 0.1 N HCl. This finding indicated that the dissolution profile of PSP could vary by changing the pH of the buffer. However, to serve the purpose of delivering this drug in the oral cavity, the release test was performed in SSF with a pH of 6.8.

Overall, owing to the higher surface area and the fast disintegration of the electrospun nanofibers (≈30 s) compared to the solvent-cast ODFs (≈700 s), the dissolution rate of PSP was accelerated, particularly at the first time points up to 15 min, which could be vital in achieving a rapid effect in specific health problems, such as allergies, asthma and inflammatory conditions. This finding can attract the attention to the use of electrospun nanofibers as potential alternatives to the conventional ODTs, especially that these fibers have shown an end-user acceptability (i.e., handling and mouthfeel to adult human participants), in a recent study by our group [31].

### 3.9. E-Tongue and Bitterness Threshold

The taste sensing system was used to quantify the sensor response for PSP as a function of its concentration, as shown in Figure 8. It was observed that the basic bitterness sensor AC0 exhibited a response to the drug. Therefore, the data from the AC0 sensor can reliably be used to assess the taste of formulations.

Quinine hydrochloride dihydrate was chosen as a bitter reference drug, and its bitterness profile is shown in Figure 9A. The study by Soto et al. [40] determined the bitterness threshold of quinine from human sensory panels to be 0.26 mM, this value being the EC_50_ or half of the maximum taste rating at which the drug was found to be aversive. This equates to an e-tongue AC0 sensor output of 117 mV when fitted on a logarithmic trend-line using the equation shown in Figure 9A.

To estimate the bitterness threshold of PSP, this value of 117 mV was used and substituted into the corresponding logarithmic equation to generate a mean drug concentration that matches quinine’s known bitter, mean sensor response [62]. The aversiveness of PSP can, therefore, be assumed to be 2.79 mM, as calculated using the equation in Figure 9B. This result indicated that PSP is approximately ten times less bitter than quinine HCl dihydrate, which contradicts the findings of Zheng and Keeney (2006) who reported that prednisolone (in a sodium salt) had a more bitter taste than quinine hydrochloride [33].

Whilst this seems like an enormous difference, quinine is a particularly bitter drug, and an e-tongue sensor response of higher than five mV does indicate bitterness and therefore, necessitates taste-masking when formulating an orally dispersible formulation. This finding suggested the need to taste-mask PSP, mainly if this drug was intended for pediatric patients. Besides, the use of e-tongue can be an artefact for a quick taste detection of drugs, which could be an alternative to the in vivo taste assessments.

## 4. Conclusions and Future Perspectives

This preliminary work demonstrated the better performance of the electrospun nanofibers compared to the ODFs, particularly in the disintegration and dissolution studies. Despite that the EE% of the ODFs was higher (≈100%) than the nanofibers (≈87%), the disintegration time was substantially quicker for the electrospun nanofibers (≈30 s) than the ODFs (≈700 s). The accelerated disintegration and release behavior of the nanofibers can attract the attention as a potential alternative to the conventional PSP ODTs. This has enhanced the dissolution rate of PSP which could, in turn, improve the drug bioavailability and therefore, achieve a rapid effect in specific health problems, such as allergies, asthma, and inflammatory conditions. It is also worth mentioning that the therapeutic dose of PSP ODT, i.e., 13.4 mg, has approximate weights of 92 and 80 mg of the drug-loaded nanofibers and ODFs, respectively, which will be enough to be administered compared to the 500 mg tablet weight of a single ODT. In addition, the drug was confirmed to be bitter, and the aversiveness concentration was quantified (2.79 mM), making it an ideal candidate to be electrospun into an oral film, as this manufacturing method has been used for taste-masking application. A coaxial (core/shell) nanofiber is suggested, in which a taste-barrier polymer Eudragit E PO can be applied as the shell for the PSP–PVA nanofibers. This newly proposed formulation should be further evaluated using the e-tongue and other in vitro and in vivo assessments to appraise their safety and efficacy more comprehensively.

## Figures and Tables

**Figure 1 pharmaceutics-13-00120-f001:**
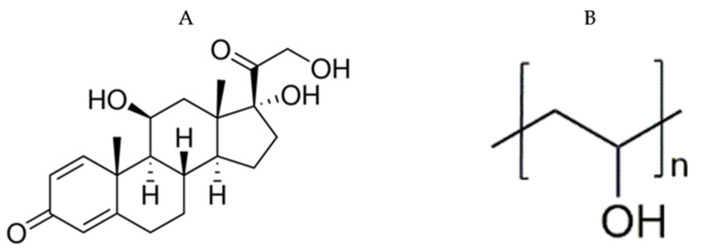
The chemical structures of: (**A**) prednisolone sodium phosphate (PSP) drug and (**B**) Polyvinyl alcohol (PVA). These structures were drawn by ChemDraw Professional 16.0.

**Figure 2 pharmaceutics-13-00120-f002:**
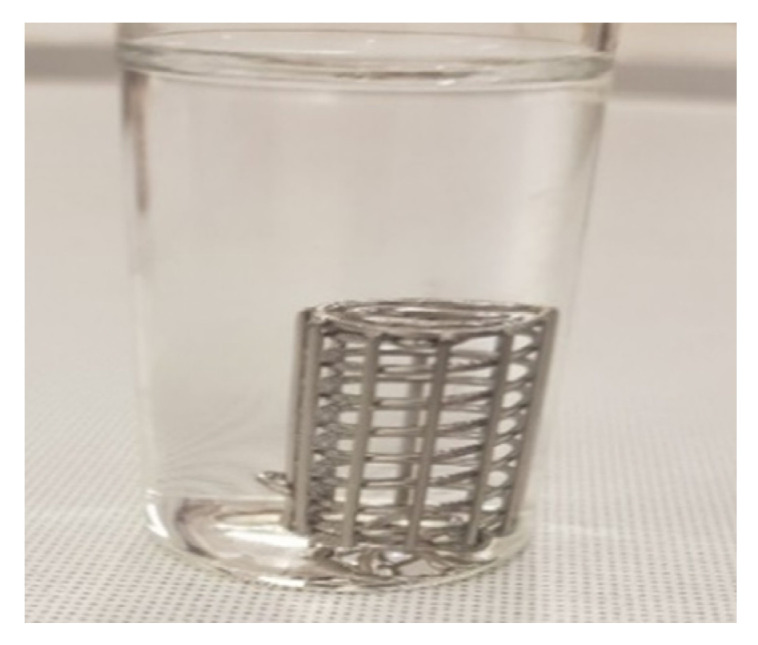
A custom-made sinker used in the release study to avoid the floating of the fibers and the oral dispersible film (ODF).

**Figure 3 pharmaceutics-13-00120-f003:**
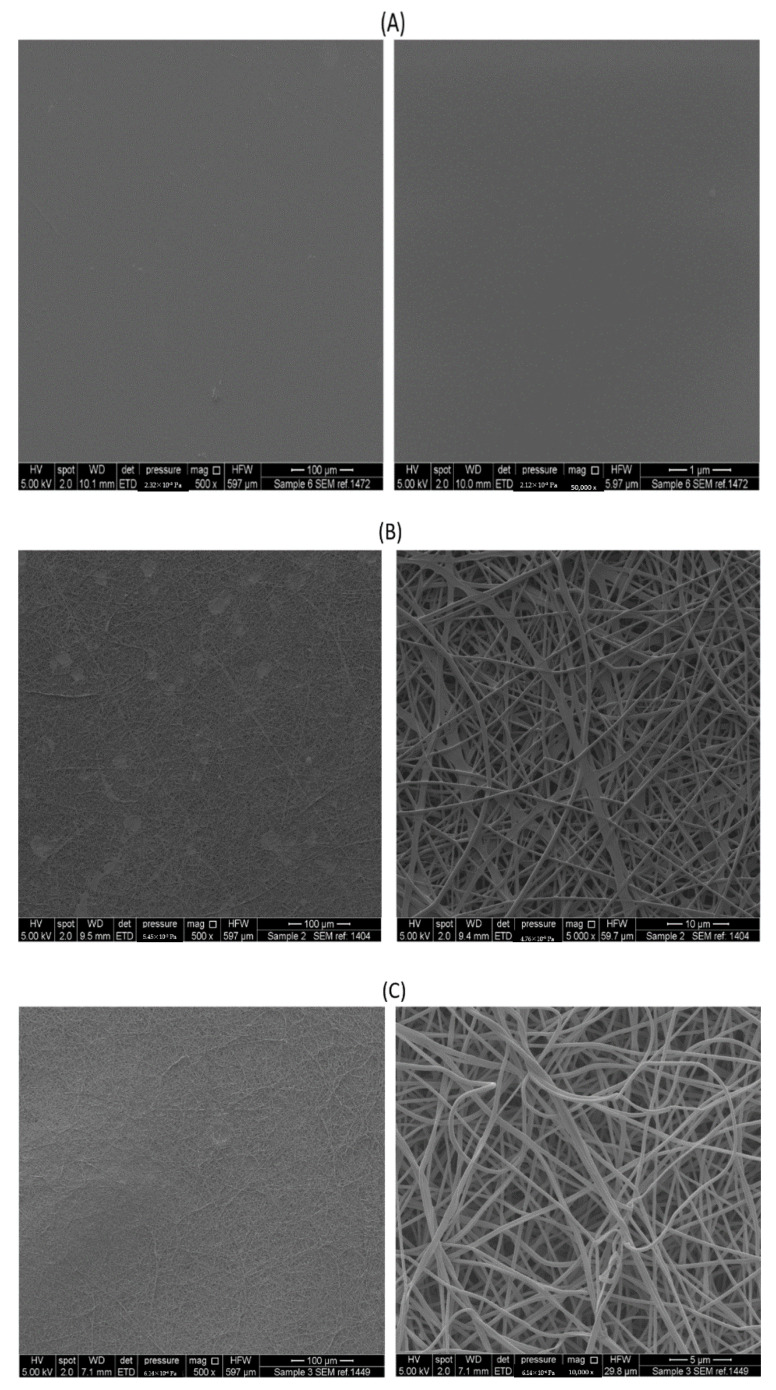
Scanning electron microscopy (SEM) images showing the morphology of: (**A**) drug-loaded ODF as a flat surface, (**B**) drug-loaded nanofibers prepared with no addition of ethanol showing a smooth, un-beaded, and un-porous, but flattened fibrous network, and (**C**) drug-loaded nanofibers prepared by the addition of 10% *v*/*v* ethanol showing a smooth, un-beaded, and un-porous fibrous network. (**D**) Nanofibers fibers size distribution of the drug-loaded nanofibers measured from 100 different nanofibers.

**Figure 4 pharmaceutics-13-00120-f004:**
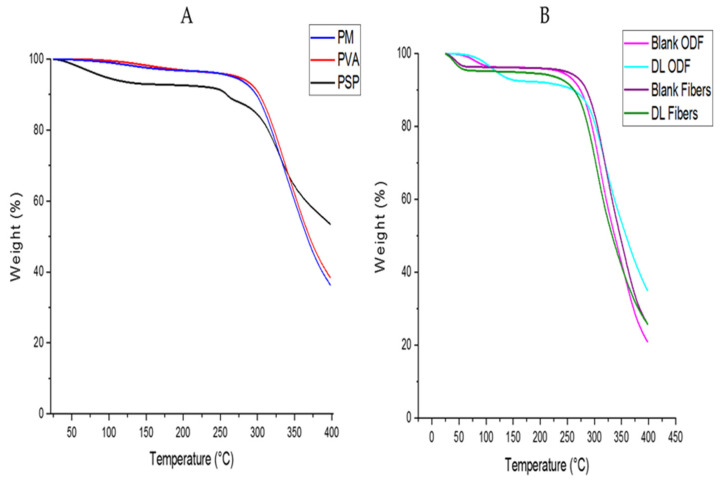
Thermogravimetric analysis (TGA) showing the decomposition temperature of (**A**); PSP, PVA and physical mixture (PM), (**B**); blank and Drug Loading (DL) nanofibers and ODFs.

**Figure 5 pharmaceutics-13-00120-f005:**
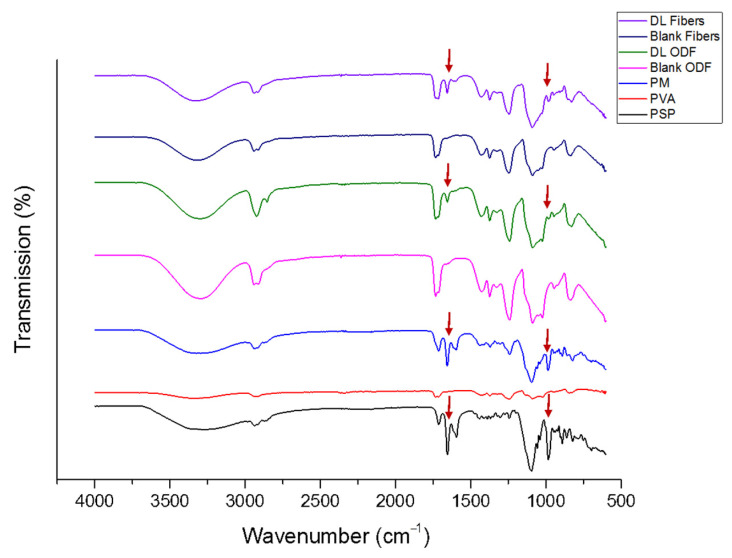
Fourier Transform Infrared (FTIR) transmissions of PSP, PVA, PM, blank and drug-loaded nanofibers, and ODFs showing the distinctive drug peaks at 1686 cm^−1^ and 1014 cm^−1^ that appear in the PM and the DL fibers and ODFs compared to the blank formulations. PM: physical mixture; DL: drug-loaded.

**Figure 6 pharmaceutics-13-00120-f006:**
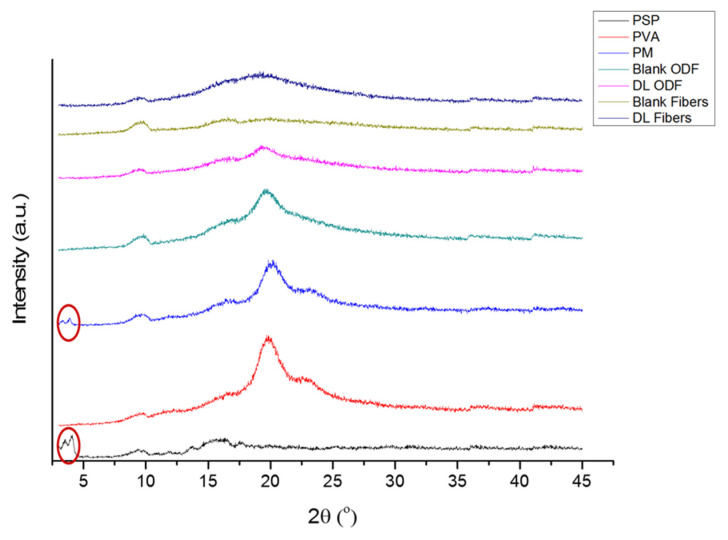
X-ray diffraction (XRD) patterns of PSP, PVA, PM, blank and drug-loaded nanofibers, and solvent-cast ODFs showing that PSP and PVP are in the crystalline form due to the presence of distinctive peaks between 3 and 5° and 18 and 21°, respectively. These peaks were also presented in the PM but not in the DL formulations, indicating the molecular dispersion of the drug within the nanofibers and ODFs. PM: physical mixture; DL: drug-loaded.

**Figure 7 pharmaceutics-13-00120-f007:**
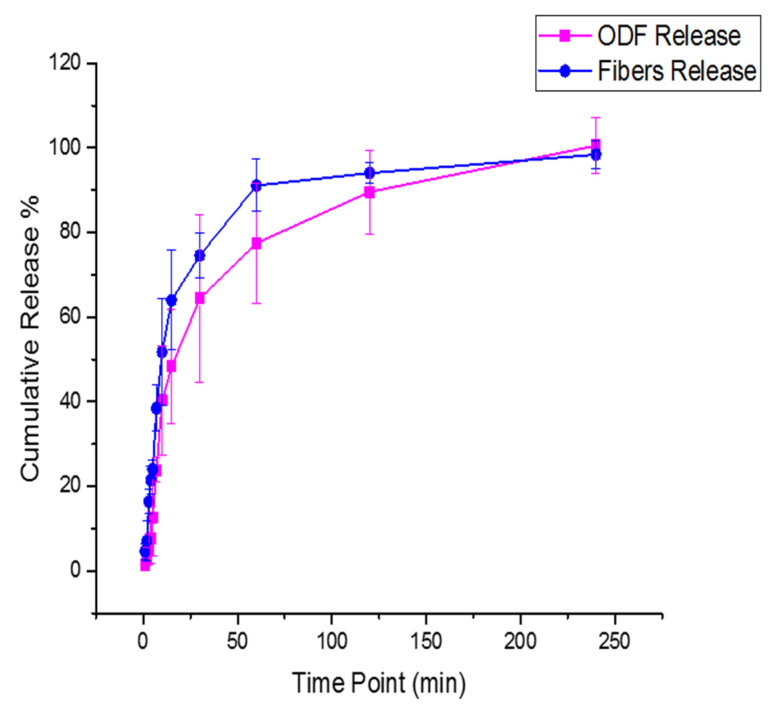
The cumulative release profile of the drug-loaded nanofibers and ODFs, showing a significantly (*p* < 0.0001) faster drug release profile of PSP-loaded nanofibers compared to PSP-loaded ODFs after 240 min in SSF (pH 6.8).

**Figure 8 pharmaceutics-13-00120-f008:**
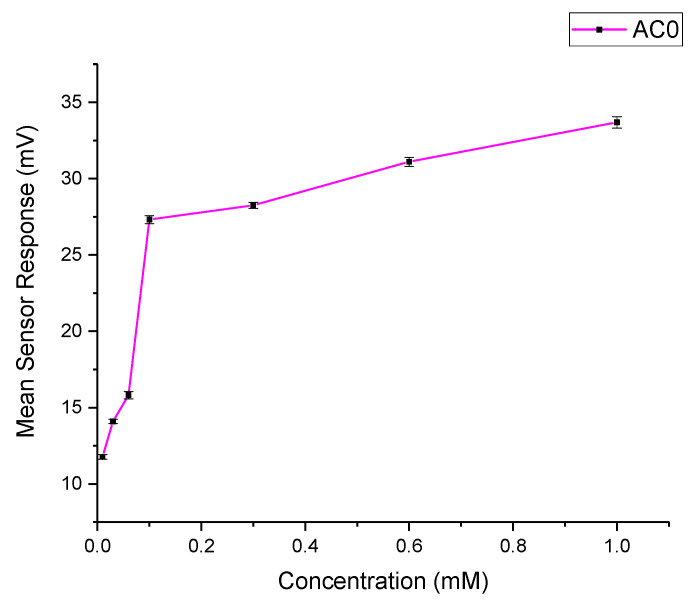
AC0 sensor response for PSP as a function of concentration.

**Figure 9 pharmaceutics-13-00120-f009:**
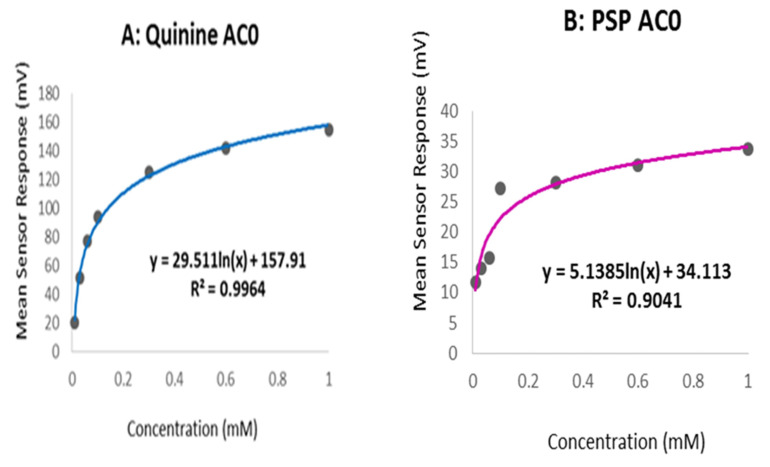
AC0 logarithmic trend-line response curve for (**A**) quinine HCl dihydrate and (**B**) PSP.

**Table 1 pharmaceutics-13-00120-t001:** DL and EE% (±SD) of the drug-loaded formulations (*n* = 6).

Formulation	DL (µg/mg)	EE%
Nanofibers	146.2 ± 1.8	87.5 ± 1.1
Solvent-cast ODFs	167.2 ± 2.4	100.1 ± 1.4

## Data Availability

All data are available upon request.

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
