# Peer review of "A Potential Alternative Orodispersible Formulation to Prednisolone Sodium Phosphate Orally Disintegrating Tablets"

_pharmaceutics, 2021, doi:10.3390/pharmaceutics13010120_

Round 1

Reviewer 1 Report

Manuscript ID: pharmaceutics-1067520

Title: A Potential Alternative Orodispersible Formulation to Prednisolone Sodium Phosphate Orally Disintegrating Tablets for the Treatment of Allergic Reactions

Authors: Essam A. Tawfik * , Mariagiovanna Scarpa , Hend E. Abdelhakim , Haitham A. Bukhary , Duncan Q. M. Craig , Susan A. Barker , Mine Orlu

In the article "A Potential Alternative Orodispersible Formulation to Prednisolone Sodium Phosphate Orally Disintegrating Tablets for the Treatment of Allergic Reactions" the authors describe the use of electrospun nanofibers and solvent-cast oral dispersible films as a potential OD formulations for prednisolone sodium phosphate that is commercially available as ODTs.

The selective comments towards the authors are:

  1. What is the reasons or the advantages of using nanofibres? The authors must demonstrate what is the novelty?

Scientifically speaking all the conducted experiments lead to the conclusion that the  electrospun nanofibers have better performance compared to the ODFs. But I feel they still need to ameliorate the paper before publishing it.

  1. page 1, line 42: a citation must be added when the authors cite the US FDA. Maybe this one:

"U.S. Department of Health and Human Services Food and Drug Administration Center for Drug Evaluation and Research (CDER) (2008) Guidance for Indus­try Orally Disintegrating Tablets - CDER Data Standards Manual. Chemistry: 1-3."

  1. Figure 2B SEM image. The authors must added a new figure at another resolution. The figure provided by the authors seems to be blank.

  1. The authors must added the method of preparation of the physical mixture .

  1. The desintegration test was observed with a camera? which was the composition of the desintegration medium? The substances were pluged into SSF at 37o C or the nanofibres and the ODFs were plunged into medium and then it was heated at 37o C? The authors must explain this process. It was used only one tablet? The experiments were repeated?

  1. The Conclusion part must be concise and written as one paragraph.

Author Response

We thank the referees for their thoughtful and helpful comments.  We respond to each point below; their suggestions have undoubtedly improved the paper, for which we are very grateful. Please find the attached word file.

Reviewer 2 Report

The manuscript entitled “A Potential Alternative Orodispersible Formulation to Prednisolone Sodium Phosphate Orally Disintegrating Tablets for the Treatment of Allergic Reactions” investigated electrospun nanofibers and solvent-cast oral dispersible films (ODFs) for prednisolone sodium phosphate.  

  1. Introduction section should be reorganized. It would be nice to explain the importance of this study, not just a list of knowledge.
  2. How do the authors normalize drug dose in electrospun nanofibers?
  3. Justify the selection of the disintegration medium (SSF, pH6.8) and volume (60 mL) for an orally disintegrating film.
  4. Author should move Figure 4 to Materials section.
  5. Figure 7 should be deleted because calibration curve is basic data. Linearity can be described in section 3.5. Ultraviolet Assay for Drug Determination.
  6. In Figure 8, is there statistical difference between nanofibers and ODF? Author should compare drug release profiles of ODF and nanofibers to commercial prednisolone ODT.

Author Response

(The authors gave the same response as above.)

Reviewer 3 Report

  1. It is not clear what means EE % in Abstract (Abstract, 22 row).  Then it is mentioned first time in abstract it should be full name of abbreviation.
  2. From Introduction part is not clear the aim of this research.
  3. The name of section “2.2.1. Nanofibers Preparation“ may be change to „Preparation of electrospun mat“.
  4. Page 4 165 row „Fiber size analysis was performed by measuring  the diameter of these fibers using ImageJ software ..“ should be mention how many electrospun fibers were measured in one sample.
  5. Page 4 171 row „drug-loaded nanofibers“ may be change to drug loaded electrsopun mat from nano-microfibers“. In 6 page 267 row it was mentioned that mean diameter of electrospun fibers is 257 ± 52 nm. From average is not clear how many were estimated fibers with diameter to 100 nm (nanofibers). Analyzing the structure of electrospun materials is better to show frequency distribution of electrospun fibers diameter.
  6. One SEM image of 5 µm at mag X50000 show only very small part of electrospun material. In order to prove that there was no of drug crystals should be analyzed SEM images of higher scale. Also it would be useful to analyze SEM images (diameter of fibers, the structure of electrospun mat (evidence of spots of polymer solution on electrospun mat)) of pure PVA without drug (blank fibers) and PVA with drug loaded (DL loaded fibers).
  7. What surface area (g/m2) of same size samples of electrospun mat and ODF mats? Electrospun mats are porous material and porosity may have influence on analyzing properties.
  8. The PVA solution for ODFs by solvent casting was prepared similar to electrospun solution, only 1 ml of ethanol was not used? In order to prove assumption of TGA „This result suggested that the nanofibers might contain a slight residual amount of ethanol“ may be analyzed ODF with ethanol also. Maybe the difference between results are due porosity of electrospun mat (not due ethanol)? It may be prove only if will be analyzed ODF also with ethanol.
  9. Page 6 , 260 row “Owing to the high molecular weight of PVA (197,000 Dalton) in the formulation, flattened fibers were obtained, which was also reported in Koski et al. study [40].” It is not correct. In this study not flattened fibers obtained. Flattened fibers are ribbon like fibers.
  10. In the name of article are mentioned allergic reactions, but there is no analysis of electrospun or ODF influence on allergic. The name of article may be changed.
  11. What size / weight of samples were used in X-Ray diffraction test?
  12. The structure of electrospun mat depends from polymer solution properties, technological properties, type of electrospinning apparatus (syringe type, needless type, rotating collector, plate collector, type of collecting material (foil, textile material) due it if in one study was estimated that was performed good electrospun material, for example, then was used ethanol, in other study may be other results only because that the type of electrospinning apparatus was different. It would be useful to analyze electrospun mat and ODF of identical composition (both samples with ethanol or without ethanol).

Author Response

(The authors gave the same response as above.)

Round 2

Reviewer 1 Report

The Authors have satisfactorily responded to all reviewer questions and made the necessary changes to the manuscript.

In consequence, I agree with the publication of the article "A Potential Alternative Orodispersible Formulation to Prednisolone Sodium Phosphate Orally Disintegrating Tablets for the Treatment of Allergic Reactions " in the present form.

Author Response

We would like to thank you for your fruitful comments that definitely improved the manuscript 

Reviewer 2 Report

Previously, reviewer commented "In Figure 8, is there statistical difference between nanofibers and ODF? Author should compare drug release profiles of ODF and nanofibers to commercial prednisolone ODT", authors response is not enough to solve the questions of potential readers and researchers.

If authors do not add the drug release profile of Orapred ODT in this manuscript, author should discuss it and add refs in Result and discussion section. Author described "A Potential Alternative Orodispersible Formulation to PSP ODTs" in the title, as alternatives, dissolution rate is the most important in the development of oral dosage forms.  

Author Response

Previously, reviewer commented "In Figure 8, is there statistical difference between nanofibers and ODF? Author should compare drug release profiles of ODF and nanofibers to commercial prednisolone ODT", authors response is not enough to solve the questions of potential readers and researchers.

Apologies for this, the significance of the release study of the ODF and nanofibers has been calculated according to the below method, which was added in the methodology section 2.2.10.

‘For the release study, the mean comparison was performed by parametric T-test using GraphPad Prism® statistical software. The P < 0.0001 was taken as a criterion for a statistically significant difference.’

After performing the above test, it was found that ‘the drug release rate of the nanofibers is significantly faster (P < 0.0001) than the ODFs’. This statement was added in lines 462 and 498.

If authors do not add the drug release profile of Orapred ODT in this manuscript, author should discuss it and add refs in Result and discussion section. Author described "A Potential Alternative Orodispersible Formulation to PSP ODTs" in the title, as alternatives, dissolution rate is the most important in the development of oral dosage forms. 

The Orapred ODT release was described in the text, due to the difficulty of performing the test.

The following section has been added in the results and discussion section 3.8.

 ‘On the other hand, the dissolution of PSP ODT (Orapred ODT®) was previously tested by Adinarayana et al. [61] in three different release media, i.e. water, 0.1N hydrochloric acid (HCl) and acetate buffer (pH 4.5). The results showed that 81% of the drug was released after 45 minutes in water, while full PSP release was obtained at 30 minutes in the acetate buffer and after 60 minutes in 0.1N HCl. This finding indicated that the dissolution profile of PSP could vary by changing the pH of the buffer. However, to serve the purpose of delivering this drug in the oral cavity, the release test was performed in SSF with a pH of 6.8.’

[61] Adinarayana, N., Raj, S.B., Rajasekhar, M., Reddy, K.B. and Mohanambal, E., 2011. Formulation and evaluation of prednisolone sodium phosphate orally disintegrating tablets. International Journal of Research in Pharmaceutical Sciences, 2(2), pp.192-199.

Reviewer 3 Report

1. One SEM image of 5 µm at mag X50000 or 1 µm show only very small part of electrospun material. In order to prove that there was no of drug crystals should be analyzed SEM images of higher scale, i.e., at 50 µm or 100 µm. It would not possible to measure diameter of fibers at 50 µm SEM images, but for overall structure analysis it is necessary SEM images at higher scale. From higher scale it is possible to notice spots of polymer solution, impurities. If it is possible I will suggest to use SEM images of 50 µm /100 µm at X xxxx magnification  and 5 or 1 µm (where diameter of fibers was measured).

  1. In histograms not count of fibres, but frequency distribution (%) is used (Figure 3) in scale Y. In scale X , it should be diameter in nm (the name of fibers is nanofibers in this study).
  2. 288 row “Owing to the high MW of PVA (197,000 Dalton) in the formulation, flattened fibers were obtained“. In this study article „Potential Alternative Orodispersible Formulation to Predni-2 solone Sodium Phosphate Orally Disintegrating Tablets“ in presented SEM images there are not flat fibers (figure 3).In answers to reviewers is mentioned „The reason that we used ethanol with the fibers, is that we wanted to lose the flattened like appearance of the fibers, which indicates for their wettability (i.e. residual water)“, so in this article there is no flat fibers.

Author Response

  1. One SEM image of 5 µm at mag X50000 or 1 µm show only very small part of electrospun material. In order to prove that there was no of drug crystals should be analyzed SEM images of higher scale, i.e., at 50 µm or 100 µm. It would not possible to measure diameter of fibers at 50 µm SEM images, but for overall structure analysis it is necessary SEM images at higher scale. From higher scale it is possible to notice spots of polymer solution, impurities. If it is possibleI will suggest to use SEM images of 50 µm /100 µm at X xxxx magnification  and 5 or 1 µm (where diameter of fibers was measured).

This has been adjusted accordingly and new SEM figures have been added to show the fibers before and after using ethanol. This is to answer the third comment.

  1. In histograms not count of fibres, but frequency distribution (%) is used (Figure 3) in scale Y. In scale X, it should be diameter in nm (the name of fibers is nanofibers in this study).

You are right. Thank you for the suggestion. This has been adjusted accordingly.

  1. 288 row “Owing to the high MW of PVA (197,000 Dalton) in the formulation, flattened fibers were obtained“. In this study article „Potential Alternative Orodispersible Formulation to Predni-2 solone Sodium Phosphate Orally Disintegrating Tablets“ in presented SEM images there are notflat fibers (figure 3).In answers to reviewers is mentioned „The reason that we used ethanol with the fibers, is that we wanted to lose the flattened like appearance of the fibers, which indicates for their wettability (i.e. residual water)“, so in this article there is no flat fibers.

Apologies for the confusion. New SEM Figures have been added to show how the fibers look like before (Figure 3B) and after using ethanol (Figure 3C).